# Application of a Micro Free-Flow Electrophoresis 3D Printed Lab-on-a-Chip for Micro-Nanoparticles Analysis

**DOI:** 10.3390/nano10071277

**Published:** 2020-06-30

**Authors:** Federica Barbaresco, Matteo Cocuzza, Candido Fabrizio Pirri, Simone Luigi Marasso

**Affiliations:** 1Chilab-Materials and Microsystems Laboratory, DISAT, Politecnico di Torino, 10034 Chivasso (Turin), Italy; federica.barbaresco@polito.it (F.B.); matteo.cocuzza@infm.polito.it (M.C.); fabrizio.pirri@polito.it (C.F.P.); 2CNR-IMEM, Parco Area delle Scienze 37a, 43124 Parma, Italy

**Keywords:** 3D printing, µFFE, concentration, microfluidics, micro- and nanoparticles, separation

## Abstract

The present work describes a novel microfluidic free-flow electrophoresis device developed by applying three-dimensional (3D) printing technology to rapid prototype a low-cost chip for micro- and nanoparticle collection and analysis. Accurate reproducibility of the device design and the integration of the inlet and outlet ports with the proper tube interconnection was achieved by the additive manufacturing process. Test prints were performed to compare the glossy and the matte type of surface finish. Analyzing the surface topography of the 3D printed device, we demonstrated how the best reproducibility was obtained with the glossy device showing a 5% accuracy. The performance of the device was demonstrated by a free-flow zone electrophoresis application on micro- and nanoparticles with different dimensions, charge surfaces and fluorescent dyes by applying different separation voltages up to 55 V. Dynamic light scattering (DLS) measurements and ultraviolet−visible spectroscopy (UV−Vis) analysis were performed on particles collected at the outlets. The percentage of particles observed at each outlet was determined in order to demonstrate the capability of the micro free-flow electrophoresis (µFFE) device to work properly in dependence of the applied electric field. In conclusion, we rapid prototyped a microfluidic device by 3D printing, which ensured micro- and nanoparticle deviation and concentration in a reduced operation volume and hence suitable for biomedical as well as pharmaceutical applications.

## 1. Introduction

Micro- and nanoparticles (M/NPs) represent a consolidated instrument in micro nanomaterials engineering for a widespread number of applications that range from drug carriers [1,2,3] to cosmetics development [4] as well as from disease therapy [5,6] to chemical analysis [7]. On the other hand, in a kind of specular negative view, M/NPs also represent a major problem as contaminants in water [8,9] and soil [10]. In this context, the development of new tools that may support the analysis of these materials is of strategic relevance for all the correlated aspects, especially if these instruments have a potential low cost and can be engineered to obtain transportable equipment. Microfluidics and lab-on-a-chip (LOCs) were developed tens of years ago [11] to satisfy the requirements of rapid, low cost and small volume analyses [12,13,14,15] along with their exploitation in point-of-care (POC) systems [16]. Recently, a valuable approach for the development and fabrication of microfluidic and LOC devices has been found in 3D printing [17,18,19] and some examples demonstrate their successful employment in the separation and analysis of M/NPs [20,21].

Few techniques allow the control of the fractionation of M/NPs [22] and among these, free-flow electrophoresis (FFE) has shown itself to be a highly versatile approach. In fact, this method is suitable for preparative scale fractionation and separation of a large range of species: solid particles, cells, organelles and complex protein mixtures, etc. [23]. In detail, FFE is a continuous and analytical separation technique used to separate chemical species, micro- and nanoparticles and biomolecules in a flowing stream according to size and charge. A thin sample stream is introduced into a planar separation channel with a buffer running in parallel: when the electric field is applied perpendicularly across the separation chamber, charged analytes deflect laterally based on their electrophoretic mobility. When operating at the microscale, thus acquiring several well-known advantages related to the scaling down of the dimensions, the microfluidic approach allows for the implementation of correspondingly scaled micro free-flow electrophoresis (µFFE) [24]. Except for the first µFFE devices, which were manufactured in silicon through a standard lithographic process [25], nowadays the majority of the µFFE devices are developed by exploiting various types of polymers through soft-lithographic or replica molding strategies [26,27,28]. So far, only one µFFE device, presented in literature, has been developed by an additive manufacturing process [29]. In the latter work, the researchers suggested how three-dimensional (3D) printing would be an excellent strategy for fast prototyping, highlighting the advantages due to the reduced manufacturing time and cost since it involved no cleanroom facilities, but their device was limited by the employment of fused deposition modeling (FDM) technology.

In this work, a µFFE LOC has been developed by applying polymeric 3D printing technology to rapid prototype a chip for M/NPs testing. The employed rapid prototyping system’s ensured main advantages are: (i) a very efficient optimization of the device design by a trial and error approach; (ii) integration of the inlet/outlet ports with the proper tube interconnections; (iii) better accuracy and resolution with respect to other printing methods [29]. In particular, a versatile solution was adopted for the inlet/outlet ports, which were directly printed with the main structure and placed on the backside of the chip, which was closed by a polymethyl methacrylate (PMMA) transparent cover. With respect to other work [29], this solution allowed for an optimized view on the microfluidics and an easy inspection through the digital microscope camera. It was demonstrated that M/NPs were moved in the confined flow according to their size/charge ratio by the applied electric field thus allowing for collection and concentration in a specific outlet. A quantification of the collected M/NPs was obtained and studied for both a single and a mixed population. The achieved results are a preliminary and encouraging starting point for the final aim of concentration and characterization of a wide variety of species and in particular for the study of exosomes (EXs), which are nanosized lipid vesicles secreted naturally in the extracellular environment by any type of cell. They play a pivotal role in intercellular communication, where they take as cargo DNA, mRNA, miRNA, lipids and short-chain peptides from one cell to another [30,31,32,33]. Thus, to achieve a clear definition of their biological functions and to evaluate EXs as potential noninvasive biomarkers, they must be isolated from biological fluids. To date, EX isolation, purification and characterization have no available standardized procedures yet. Commonly, they are purified by ultracentrifugation methods or commercial kits requiring high cost and time-consuming processes, providing low efficiency and purity of the final product [34,35]. In recent years, microfluidic systems have been developed to minimize size and cost, thus reducing sample volumes, reaction time by simultaneously performing multiple steps and granting high throughputs [36,37,38]. Proof of concept experiments were successfully carried on biovesicles, namely exosomes from biological samples, and the preliminary results were analyzed and discussed.

## 2. Materials and Methods 

### 2.1. Design

The µFFE device was designed by 3D computer aided design (CAD) software (Rhinoceros 5 for MAC, McNeel Europe, Barcelona, Spain), Figure 1, and exported in STL format for the 3D printing process. The device dimensions were 40 mm in width, 85 mm in length with a height of 2.8 mm. It was characterized by a separation chamber, two electrode channels, an inlet for the buffer solution, one for the analyte solution and five outlets (Figure 1a). The separation chamber dimensions were 30 mm in length, 13 mm in width and 100 µm in height, and it was characterized by elliptic pillars with dimensions of 1 mm × 0.8 mm with a height of 100 µm. The pillars array performed both the function of structural columns (to avoid the collapse of the top cover) and of promoting the chamber filling [39]. In order to prevent electrolysis bubbles formation in the separation chamber when an electric field was applied, partition bars [28,39], with a height of 50 µm, were designed to confine the electrode channels. (Figure 1b). These constrictions hindered bubble migration from the main chamber to the electrode channels guaranteeing the stream stability during the experiment. 

Inlets and outlets had a diameter of 3.2 mm and they were characterized by channels 500 µm deep. Electrode channels were designed with the same dimensions of inlets and outlets to facilitate the electrode wire insertion. A 100 μm deep and 1.5 mm wide trench was added around the device to define the bonding area and prevented the microfluidics clogging as reported as follows. The interconnection made by the ¼’’ 28 UNF threaded fluidic fittings (Figure 1a,c) was directly integrated into the 3D design, on the bottom side of microfluidics corresponding to the inlets and outlets of the device to ensure a stable connection with the polyurethane (PU) tubes (from SMC, Tokyo, Japan) connected with standard OminiLok fittings (from Omnifit Labware, Cambridge, UK). 

### 2.2. Fabrication

The microfluidic of the µFFE device was fabricated by CAD model 3D printing (Figure 2a). The file in STL format was preprocessed by an InkJet 3D printing system (Objet 30, Stratasys, Rehovot, Isareal) using Verowhiteplus RGD835 resin and exploiting both the “glossy” and the “matte” features. After that, a 750 μm thick polymethyl methacrylate (PMMA, Evonik Industries, Essen, Germany) cover was used to seal the microfluidic chip. In detail, the top and the bottom of the device were washed by isopropanol (Merck, Darmstadt, Germany) and flushed by a nitrogen flux before sealing, then, to achieve a uniform and irreversible bonding, the 3D printed part was sealed with the cover using as a glue the Poly(ethylene glycol) diacrylate (PEGDA) 575 (Merck, Darmstadt, Germany) mixed with 1% IRGACURE 819 (Merck, Darmstadt, Germany). The bonding was achieved by clamping the whole structure inside an aluminum frame and baking for 20 min on a hot plate at 120 °C to obtain the full curing of the resin. Finally, two stainless steel wire electrodes were manually inserted in the corresponding lateral channels (Figure 2b).

### 2.3. Device Characterization and Experiment Setup 

The topography of the 3D printed devices, i.e., the features dimensions (heights, lengths and widths) and the device roughness, was characterized in order to evaluate the accuracy of the 3D printing process. For this purpose, a surface profilometer (Tencor P-10, KLA Corporation, Milpitas, CA, USA) and an optical microscope (LEICA DVM 2500, Leica Microsystems Srl, Buccinasco, Italy) were employed. The leakage and M/NPs tests were performed using a pumping system (HARVARD APPARATUS Pump33, Hollinston, MA, USA). Inlets were connected via PU tubes (OD = 2.0 mm, ID = 1.2 mm) to plastic syringes (2.5 mL volume, Terumo, Rome, Italy) to drive the flow rates of buffer and analyte solutions. Stainless steel wire electrodes (from The Crazy Wire Company, Warrington, UK) were connected to a DC power supply (C^4^D HV 230 Sequencer, Marciana, Italy) to apply the electrical field in the separation chamber during the experiments (Figure 3): the left electrode of the device was grounded while a range of positive potentials was applied at the right electrode. M/NPs tests were performed using a buffer solution of 4-(2-hydroxyethyl)piperazine-1-ethanesulfonic acid (HEPES) 20 mM at a pH of 7.5 (Merck, Darmstadt, DE) and as analyte solution HEPES 20 mM containing: 4 μm fluorescent sulfated polystyrene microparticles (from now on called MPs, from Thermo Fisher Scientific Inc., Monza, Italy) and 500 nm fluorescent carboxylated polystyrene nanoparticles (from now on called NPs, from from Magsphere Inc., Pasadena, CA, USA). The MPs maximum concentration solution (initial 100% concentrated MPs) was equal to 5.68 × 10^7^ particles/mL, while NPs maximum concentration solution (initial 100% concentrated NPs) was equal to 3.64 × 10^8^ particles/mL. The M/NPs collected at the outlets were characterized by dynamic light scattering (DLS, Malvern instrument, Malvern, Worcestershire, UK) both concerning size and zeta potential (see Appendix A). In particular, the multiple narrows mode was applied as an analysis model for the determination of the particle size, while a Cholesky model was applied for the zeta potential. M/NPs concentration at each outlet was quantified by ultraviolet−visible spectrophotometry (UV–Vis) (Multiskan FC Microplate Photometer, Thermo Fisher Scientific Inc., Monza, Italy) (see Appendix A). This technique is based on the Beer−Lambert law, which directly correlates the absorbance of the solution and the concentration of the M/NPs. To provide a correct quantification of particles concentration, calibration linear curves were performed for each fluorescent wavelength (see Appendix A). In detail, as reported from the MPs datasheet, they were analyzed with an excitation wavelength equal to 505 nm. Meanwhile, since for the NPs the producer did not report the excitation wavelength, which was 297 nm (see Appendix A), before performing the calibration curve it was necessary to characterize them by an absorbance spectrum using the UV−Vis instrument.

For the μFFE experiments dedicated to biovesicles, the employed EXs were collected from fetal bovine serum (FBS, Merck, Darmstadt, Germany) through an overnight centrifugation process at 10^5^
*g* at 4 °C. EXs and EX/NPs samples were characterized by nanoparticle tracking analysis (NTA, Malvern Panalytical instrument, Malvern, UK). Using this, particle size distributions and concentrations were determined exploiting the nanoparticles’ Brownian motion in a liquid suspension lightened by a laser light source.

For the tests with M/NPs and EXs, a volume of 60 μL (100% concentrated solution) was loaded into a 555 μL volume of buffer solution. Between each run, the device was rinsed with the buffer HEPES solution at 40 µL/min for 10 min. Finally, the percentage of particles observed of the μFFE device was determined as the ratio of percentage of M/NPs concentrations at each outlet to the total M/NPs concentrations collected at all five outlets [40].

## 3. Results and Discussion

### 3.1. Characterization of the Device Processes

Firstly, the µFFE microfluidic device was characterized evaluating the minimum feature sizes and the dimensions with respect to the CAD design. In particular, the smoothness of the walls on the 3D printed µFFE devices with respect to the CAD design was characterized and a comparison between the XY and Z resolutions was performed. 

The Object30 printer allows the building of 3D printed objects with two different types of surface finish: glossy or matte. The matte surface option introduces a complete cover and UV curing of the structural material with the support material, while the glossy one exploits only the necessary support material, ensuring cheaper and faster manufacturing and a smoother surface finishing of the microfluidic internal surfaces. Here, a comparison between the two surface finish types was reported evaluating both surface roughness and feature size. Values and error limits reported in Table 1 are defined by mediating four different printed devices of each type of surface finish, where on each device fifteen independent measurements have been taken. 

As depicted in Table 1, a relevant roughness was observed in the matte device, indeed this kind of surface finish has a roughness ten times greater with respect to the glossy device. It was also possible to appreciate the different roughness between the two surface finishes by optical microscope image (Figure 4). In fact, due to the huge roughness in the matte device, it was difficult to distinguish pillars and partition bars (Figure 4c,d). Moreover, the greater roughness induced a higher variability concerning the reproducibility of features in the matte device. Indeed, comparing the CAD design dimensions and the 3D printed device ones, it was possible to notice how the glossy surface definition results were more accurate with respect to the results from the matte one. These observations were investigated by the optical microscope images (Figure 4).

In detail, in the glossy type device, the pillar dimensions resulted in being larger with respect to CAD values of about 30 and 100 μm in length and width respectively, while in the matte device they were narrower by hundreds of microns with respect to nominal values. Concerning pillar heights, both types of surface finish reproduced accurately, the CAD values showing a discrepancy of 1 µm for the glossy type and 3 μm for the matte one. Inlet and outlet features presented an emphasized reproducibility with respect to the CAD dimensions depending on the surface finish type used. In fact, while these features in the glossy device were printed with a 4% accuracy with respect to the CAD dimensions, with the matte option this value corresponded to 21%, thus five times greater. Only concerning the partition bars, did the matte devices best fit the CAD values with respect to the glossy one, indeed the first surface finish type differed by 24 μm with respect to CAD values, while the glossy features reproduced the CAD dimensions with an error of 42 μm. 

The glossy µFFE device reported in this work, comparing CAD dimensions with the 3D printed ones, presented a better accuracy and final resolution with respect to the previously published one fabricated with the FDM printer [29]. Finally, glossy devices were employed for the separation tests.

### 3.2. Flow Confinement Test

The evaluation of the optimal flow rate value for the buffer flow confinement was determined by using HEPES 20 mM as buffer solution and an orange food dye diluted in HEPES 20 mM as analyte solution. At the beginning, analyte and buffer flow rates were imposed equal, then the buffer flow rate was doubled with respect to the analyte one. The buffer confinement was verified both at 1:1 and at a 2:1 buffer:analyte flow rate ratio (Figure 5). In order to perform µFFE tests within 10 min, the best performance for the buffer confinement was verified at 20 μL/min for the buffer solution and 10 μL/min for the analyte one, then these values were used as flow rates for the experiments presented in the following paragraph.

### 3.3. M/NPs Tests

The M/NPs and EXs were characterized by DLS concerning both size and zeta potential (Table 2, see Appendix A for electrophoretic mobility and conductivity measurements). For each set of tested analytes, the experiments in the μFFE device were repeated three times, error bars were reported according to the acquired data over the repetitions. The MPs, diluted in HEPES 20 mM, were inserted in the μFFE device at the analyte inlet and they were successfully deviated by the application of the electric field. The applied voltage (ΔV) was optimized and the solutions at the different outlets were collected and characterized by the DLS. In addition, the UV−Vis analysis allowed for the quantifying of the M/NPs concentration by means of the previously described linear calibration curve (*R*^2^ = 0.995). Figure 6a shows MP percentage of particles observed at each outlet at a defined ΔV. Increasing the ΔV the percentage of particles observed was higher for the external outlet (outlet#1) and, hence, an increasing number of MPs were collected at a further distance with respect to outlet#3. Similar trends were obtained also for the mixed population of M/NPs. In detail, at ΔV = 0, MPs were all collected in the central outlet (outlet#3), as expected, while for voltages equal or higher than 40 V they were mostly found in the external outlet (outlet#1) near the +V electrode. MPs spreading from the central outlet (outlet#3) to the external one (outlet#1) were detected at 30 V. Indeed, in correspondence of this voltage value, MPs were found in the three outlets (outlet#3, outlet#2, outlet#1) at different concentrations. These results were confirmed by DLS measurements (see Appendix A).

As in the previous characterization, in order to quantify the M/NPs outlet concentrations, the UV−Vis calibration curves were used (see Appendix A, *R*^2^ = 0.977 for NPs calibration curve). Figure 6b reports the final distribution of the M/NPs by the different applied voltage/outlets. At a lower voltage value of 30 V, M/NPs were mostly collected both in the external outlet (outlet#1) and in the central one (outlet#3). A partial shift of particle collection has been observed at 35 V, where particles were mostly found in the outlets near the +V electrode (outlet#1 and outlet#2). Finally, at 40 V, the majority of particles, 79% for MPs and 100% concerning the NPs, were collected in the external outlet (outlet#1).

A further proof of the μFFE device performance was given by calculating the collected number of M/NPs at outlet#1 when different voltages were applied to the electrodes. Considering the exact residual volume at outlet#1 and on the basis of the calibration curve, we could determine the collected number of particles (Figure 7). In detail, the number of M/NPs increased proportionally with the applied voltage at the electrodes up to 40 V. At a higher voltage (55 V) a drift toward the electrode channel occurred.

This was determined by comparing the number of M/NPs collected at outlet#1 when 55 V were applied at the electrodes to the number of M/NPs of the initial 100% concentrated solution normalized by each specific particle calibration curve. Results reported this ratio to be equal to 87% for MPs and 84% for NPs. This is because a little amount of particles passed over the partition bars toward the +V electrode channel. 

These results demonstrated the ability to modulate the percentage of particles observed as a function of the applied electric field. In addition, considering the amount of M/NPs diluted in the buffer solution during the device loading, it was possible to consider and evaluate the reconcentration of M/NPs by calculating the number of particles at the outlets with respect to the collected volume. A maximum 8.3× reconcentration factor of the M/NPs was obtained at 40 V. These are encouraging results that demonstrate the versatility of the device to tune the concentration of a M/NP population at a specific outlet in a reduced microliter range volume. 

The μFFE biological tests were performed with EXs and a mixed population of EXs and NPs. Figure 8 shows the percentage of EXs observed at the different applied voltage/outlets. At lower voltage values of 30 V and 35 V the 89% and 80% of EXs were mostly collected at the outlet#2. Then a notable shift of EX collection was observed at 40 V, where the 33% and 45% of EXs were found in the outlet near the +V electrode (outlet#1) and in the central outlet (outlet#3), respectively. A possible reason of why EXs at 40 V were located in the central outlet (outlet#3) could be given by the fact that this voltage can damage EXs thus inducing a variation in their properties. In fact, the NTA EXs analysis showed an EXs diameter expansion of about 40 nm at 40 V. Structural changes or degradations of EXs have been already observed in other isolation methods such as ultracentrifugation or depending on their storage and working temperature conditions [41]. 

Figure 9 reports the EX and NP percentages observed when 30 V were applied at the electrode. In detail, the majority of EXs (53%) were collected in outlet#2 while the majority of NPs observed (45%) were found in the outlet near the +V electrode (outlet#1), thus demonstrating the μFFE performance both with synthetic and biological samples.

## 4. Conclusions

In this work we evaluated the manufacturing and the particle deviation performance of a μFFE 3D printed LOC. The 3D printed microfluidic device was characterized by threaded fluidic fittings integrated on the back, corresponding to the inlet and outlet accesses. Features were reproduced with a 5% accuracy related to the CAD dimensions. Moreover, 3D printing allows the exploitation of low-cost materials and the easy introduction of further design updates with limited re-work time. 

A novel approach was proposed for the μFFE performance assessment using M/NPs with different dimensions, surface charges and fluorescent dyes. Following this, it was possible to quantify the M/NPs deviation efficiency at each outlet, when a different voltage at the electrodes was applied, by exploiting UV−Vis and DLS techniques. These propaedeutic experiments allowed for the successful conduct of biovehicle tests using FBS EXs. Finally, we demonstrated the possibility to tune the concentration of a population of different species of MPs, NPs, EXs in a specific outlet at a defined voltage and also to accumulate them in a microliter volume range. Since the μFFE layout presents a limited number of outlets, further work will be focused on redesign, the investigation of fine ΔV tuning, testing with different solutions and pH buffers to achieve optimization of particle separation and concentration.

## Figures and Tables

**Figure 1 nanomaterials-10-01277-f001:**
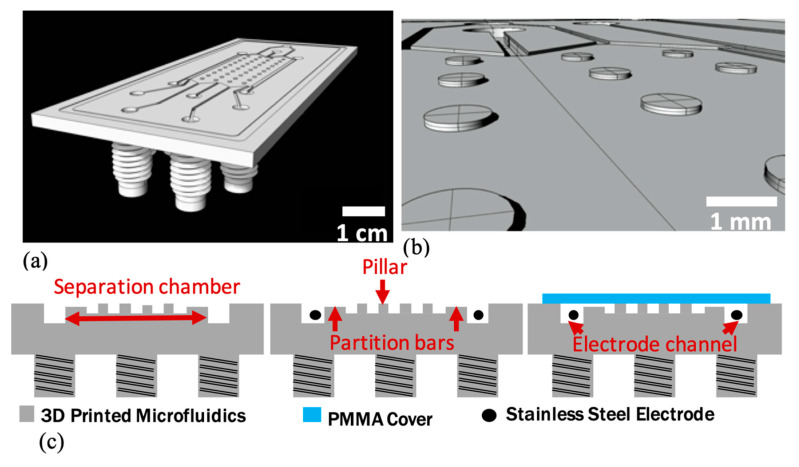
Chip layout: (**a**) the chip was composed of a separation chamber (30 mm long, 13 mm wide, 100 m high), two electrodes channels 500 μm deep, two inlets and five outlets with 3.2 mm of diameter. (**b**) Inset of the separation chamber: elliptic pillar arrays 1 mm × 0.8 mm × 0.1 mm and partition bars 50 μm deep between the separation chamber and the electrode channels. (**c**) Fabrication process flow: 3D printed microfluidics, electrode insertion and device bonding.

**Figure 2 nanomaterials-10-01277-f002:**
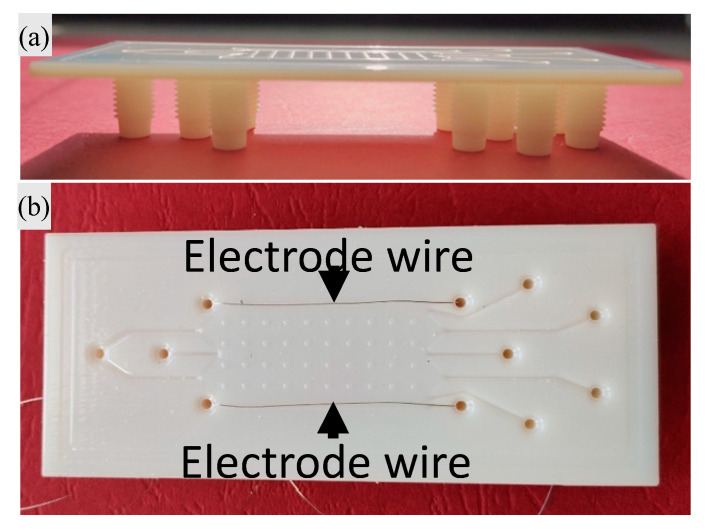
Micro free-flow electrophoresis (µFFE) device fabrication: (**a**) 3D printed microfluidics and (**b**) electrode insertion in the final device.

**Figure 3 nanomaterials-10-01277-f003:**
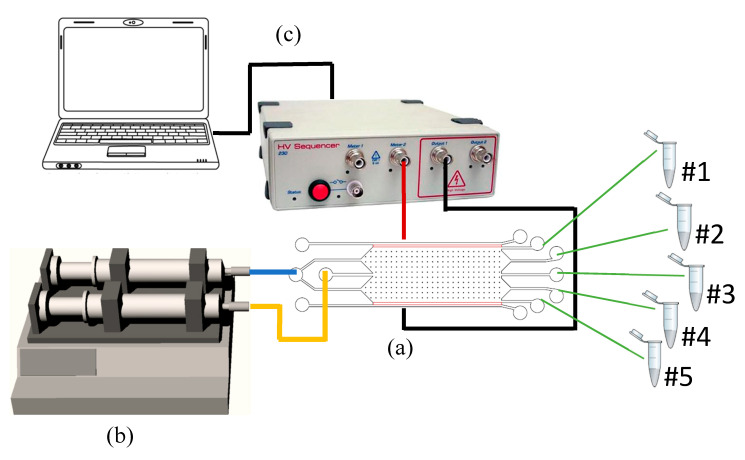
Experimental setup: (**a**) µFFE device sketch in which the yellow line indicates the solution containing the analyte, the blue one the buffer solution, red is the +V and black the ground electrode, the green lines are the sample outlets; (**b**) the syringe pump system and (**c**) the power supply.

**Figure 4 nanomaterials-10-01277-f004:**
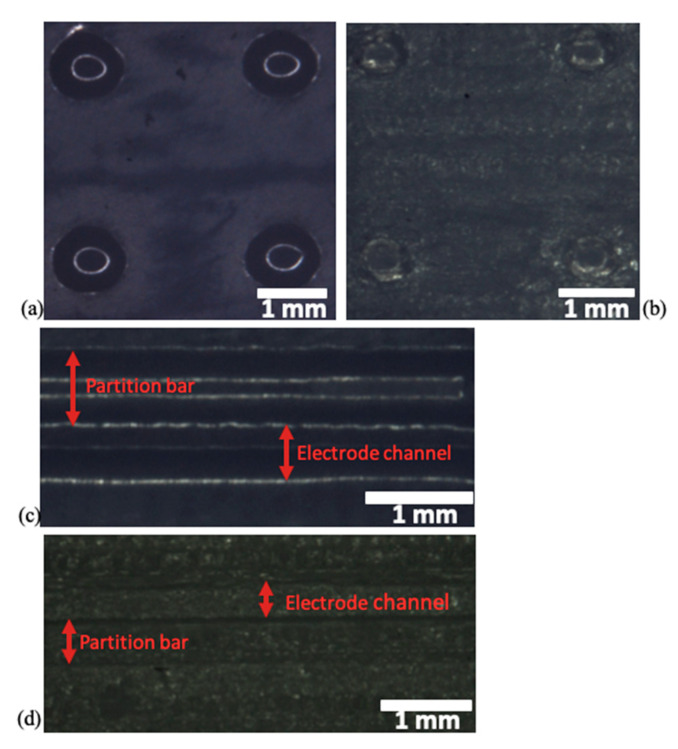
Optical microscope images: (**a**) pillars in the glossy surface device, (**b**) pillars in the matte surface device, (**c**) partition bar and electrode channel in the glossy surface device and (**d**) partition bar and electrode channel in the matte surface device.

**Figure 5 nanomaterials-10-01277-f005:**
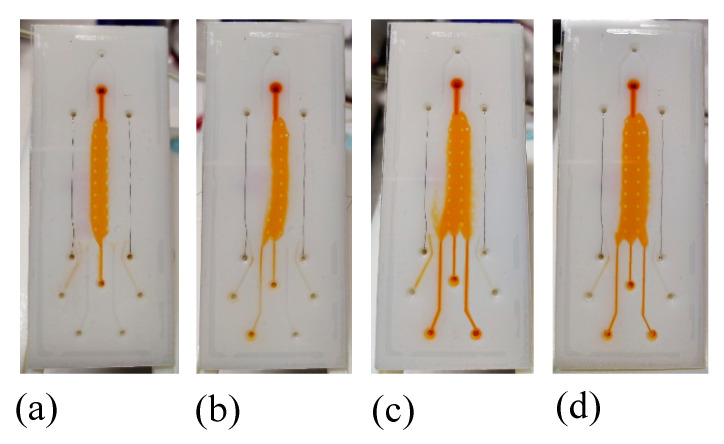
Optical photograph of the buffer flow confinement into the separation chamber: (**a**) buffer flow rate 20 μL/min and analyte flow rate 10 μL/min, (**b**) buffer flow rate 10 μL/min and analyte flow rate 10 μL/min, (**c**) buffer flow rate 10 μL/min and analyte flow rate 5 μL/min and (**d**) buffer flow rate 5 μL/min and analyte flow rate 5 μL/min.

**Figure 6 nanomaterials-10-01277-f006:**
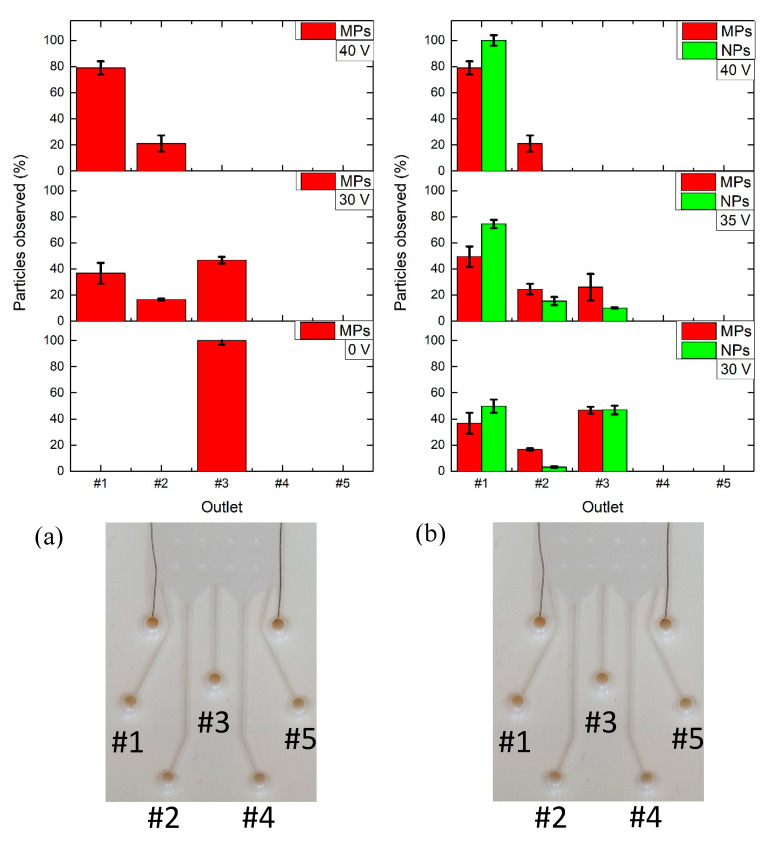
Micro- and nanoparticle (M/NP) collections at the outlets of the glossy µFFE device by setting a buffer flow rate of 20 μL/min and an analyte flow rate of 10 μL/min for 10 min: (**a**) MP outlet collection for ΔV = 0, 30, 40V, (**b**) M/NP outlet collection for ΔV = 30, 35, 40V. A percentage of particles observed, 100% and 79%, for MPs and for NPs, respectively, was obtained at ΔV = 40V. The corresponding outlets are reported in the inset for both (**a**) and (**b**).

**Figure 7 nanomaterials-10-01277-f007:**
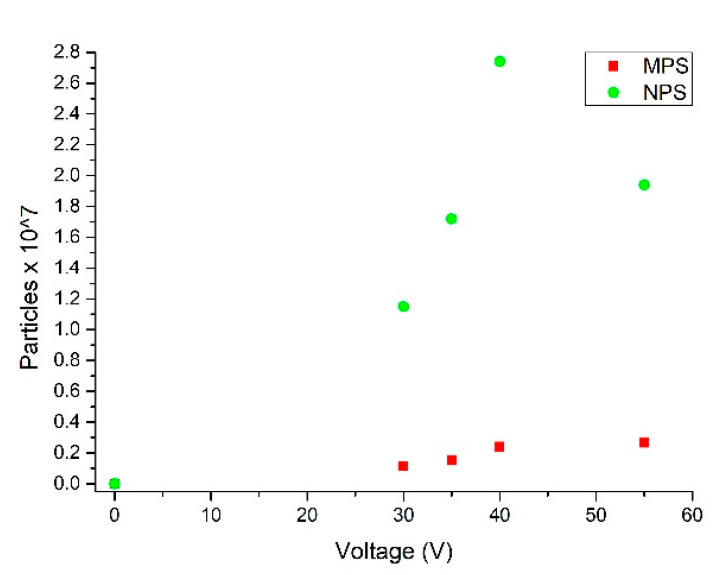
Number of M/NPs collected at outlet#1 when ΔV = 0, 30, 35, 40, 55 V were applied at the electrodes.

**Figure 8 nanomaterials-10-01277-f008:**
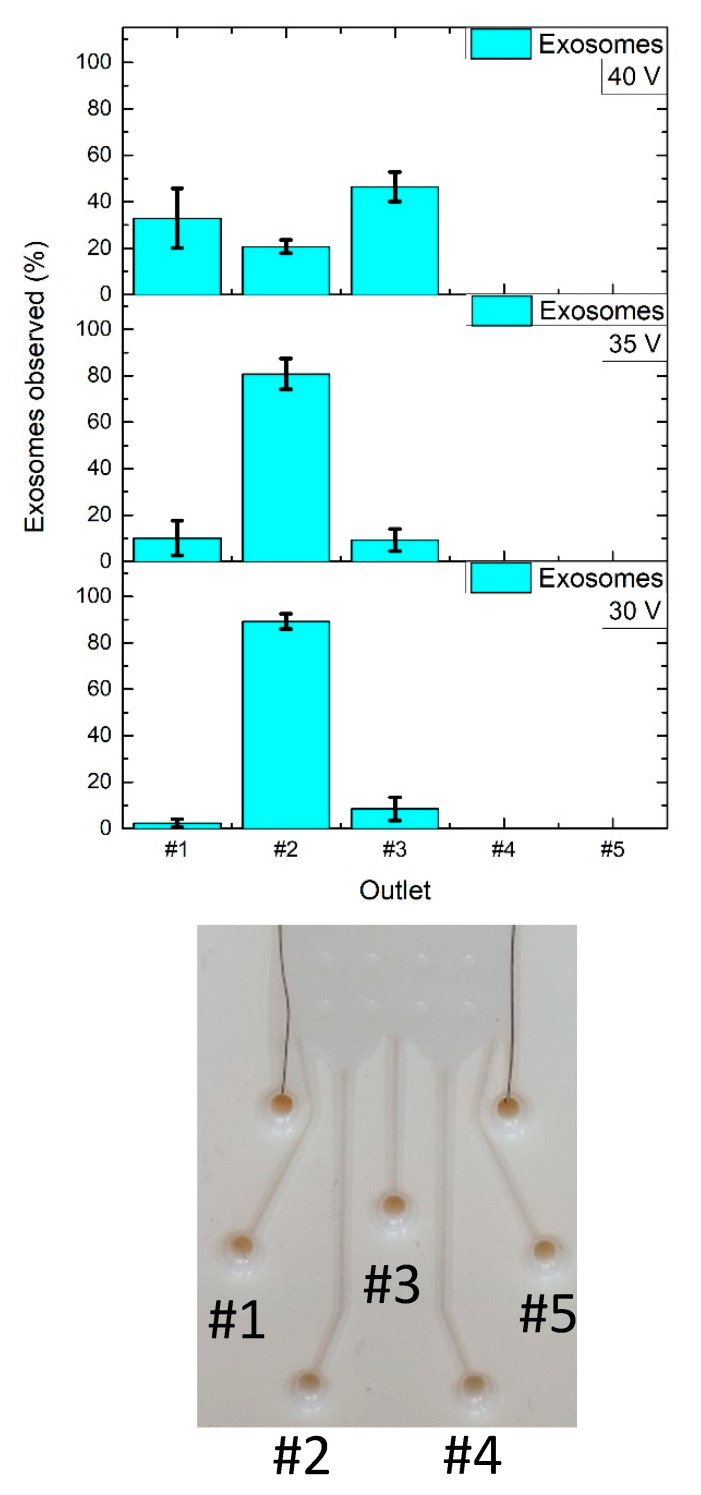
Exosome (EX) collections at the outlets of the glossy µFFE device by setting a buffer flow rate of 20 μL/min and an analyte flow rate of 10 μL/min for 10 min when ΔV = 30, 35 and 40 V were applied at the electrodes.

**Figure 9 nanomaterials-10-01277-f009:**
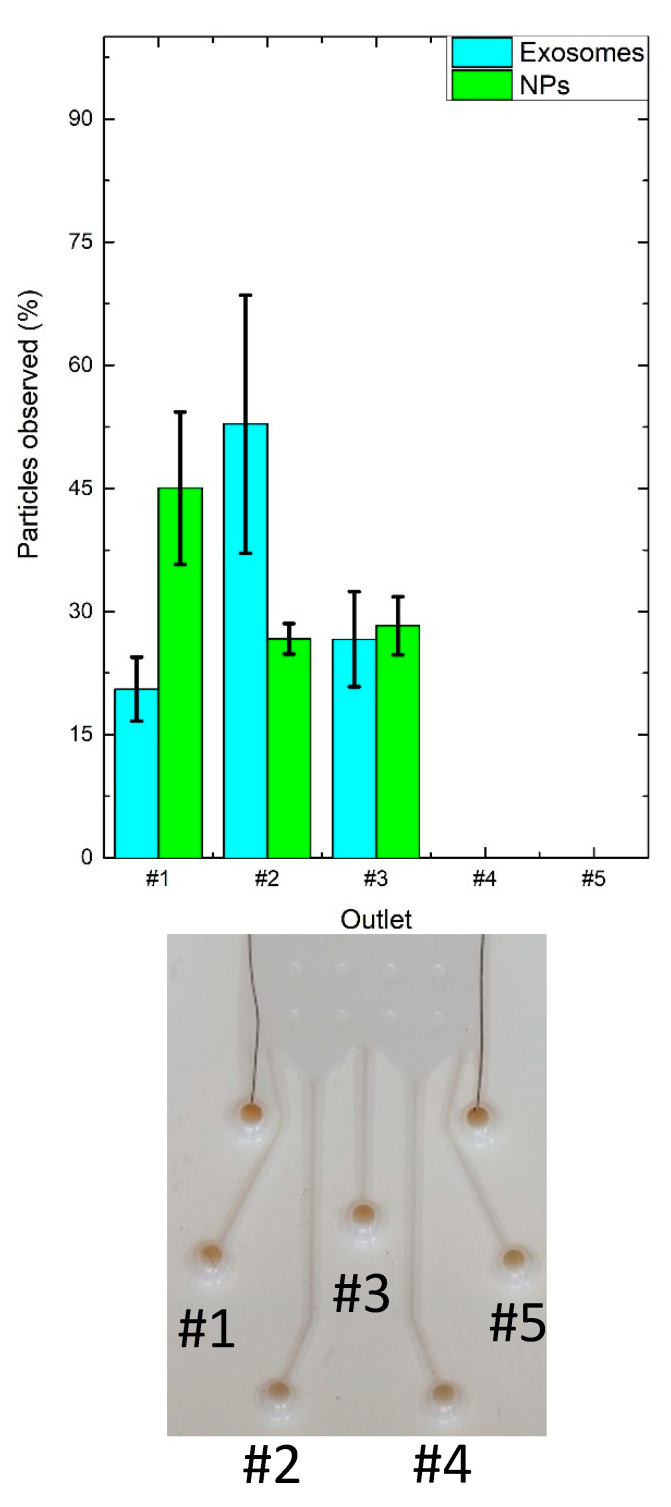
EX and NP collections at the outlets of the glossy µFFE device by setting a buffer flow rate of 20 μL/min and an analyte flow rate of 10 μL/min for 10 min when ΔV = 30 V were applied at the electrodes.

**Table 1 nanomaterials-10-01277-t001:** CAD values versus real values.

Features	CAD (µm)	Glossy Surface (µm)	Matte Surface (µm)
Pillar length	1000	1032 ± 13	725 ± 56
Pillar width	800	933 ± 23	603 ± 33
Pillar height	100	99 ± 2	103 ± 3
Inlet/outlet diameter	3200	3090 ± 195	2519 ± 81
Electrode channel width	500	612 ± 31	577 ± 33
Partition bar width	500	458 ± 15	524 ± 30
Partition bar height	50	55 ± 2	52 ± 8

Uncertainties are standard deviations of 15 measurements made on 4 printed chips.

**Table 2 nanomaterials-10-01277-t002:** Dynamic light scattering (DLS) particle measurement.

Sample	Size (μm)	Zeta Potential (mV)
MPs	3.9 ± 0.2	−10 ± 3
NPs	0.5 ± 0.05	−21 ± 5
EXs	0.056 ± 0.02	−9.61 ± 2

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
