# Peer review of "Application of a Micro Free-Flow Electrophoresis 3D Printed Lab-on-a-Chip for Micro-Nanoparticles Analysis"

_nanomaterials, 2020, doi:10.3390/nano10071277_

Round 1
Reviewer 1 Report
In this study, Barbaresco et al. developed a prototyped microfluidic device by 3D printing and used it for micro and nanoparticles separation analysis. The authors optimized the device design and operation parameters (such as flow rates and voltages), and examined the separation performance of 4 micrometers and 500 nanometers polystyrene particles. Generally, this study provided an alternative strategy for fabrication of portable device for particle separation. However, there are still some major problems that need to be addressed. A major revision is needed before it can be published.
Major comments:
1) Microfluidic technique (including 3D printing) for particle separation is not a new topic. The authors need to present the current progress and the unique features of their technique in this study.
2) In the last paragraph of the introduction section, the authors used pretty much content to outlook the application capability of their technique in exosome analysis, but they did not do any biological analysis in this study.
3) The optimization process of pillar size needs to be provided.
4) In figure 5, the full optical photograph of chamber device is suggested to be provided for better presentation of the separation performance from inlet to outlet.
5) In section 3.2, as for the flow confinement test, the flow rate range tested is pretty narrow, the authors need to examine more higher flow rates to demonstrate the rapid analysis.
6) In this study, the authors used a relative simple model: only two different-sized polystyrene particles with similar surface charge. However, the situation is indeed more complicated in real biological settings. To make this study more attractive, it is recommended to add one test using at least three different-sized micro/nanoparticles or different surface-charged particles or real biological samples.
7) Lines 70-81, the authors mentioned the separation of exosomes, but they did not provide any direct tests. If the authors can demonstrate the separation of biological objects (e.g., cells and exosomes) using their 3D-printed electrophoresis device, this study will be very interesting.
Minor comments:
1) Line 100, “fittings (Fig. 1a,c). was directly”, correct this sentence.
2) Lines 155 and 156, “a volume of 60 L (100% concentrated solution) was loaded in a 555 L volume”, clarify the unit of volume.
Author Response
1) Microfluidic technique (including 3D printing) for particle separation is not a new topic. The authors need to present the current progress and the unique features of their technique in this study.
R1.1: we want to thank the referee for the remarks. The use of 3D printing and M/NPs separation are propaedeutic for bio-vesicles separation, our final goal, which has been never attempted by other researchers in these conditions. Experiments with bio vesicles has now been added in the Materials and Methods, Results and Discussion and Conclusion sections of the manuscript. The differences from previous works have been highlighted in accordance with the major results obtained.
In the introduction the text was modify with additional information and revised properly as follows:
Line 63-72: “In this work, a µFFE LOC has been developed by applying polymeric 3D printing technology to rapid prototype a chip for M/NPs testing. The employed rapid prototyping system ensured main advantages which are: i) a very efficient optimization of the device design by trial and errors approach; ii) integration of the inlet/outlet ports with the proper tubes interconnection; iii) a better accuracy and resolution with respect to other printing methods [29]. In particular, a versatile solution was adopted for the inlets/outlets ports, which are directly printed with the main structure and placed on the backside of the chip, which was closed by a PolymethylMetacrylate (PMMA) transparent cover. With respect to other work [29], this solution allows for an optimized view on the microfluidics and an easy inspection through the digital microscope camera.”
Line 86-88: “Proof of concept experiments were successfully carried on bio vesicles, namely exosomes from biological sample, and the preliminary results were analyzed and discussed.”
2) In the last paragraph of the introduction section, the authors used pretty much content to outlook the application capability of their technique in exosome analysis, but they did not do any biological analysis in this study.
R1.2: We want to thank the referee for the remarks. As anticipated in the previous answer, the use of 3D printing and M/NPs separation are propaedeutic for bio-vesicles separation, our final goal, which has been never attempted by other researchers using FFE microfluidics in our conditions. Experimental part with bio vesicles has now been added in the Materials and Methods, Results and Discussion and Conclusion sections of the manuscript. In particular:
Lines 163-167: report the materials and methods on the EXs experiments.
Lines 287-301: report biological tests performed with EXs and a mixed population of EXs and NPs.
3) The optimization process of pillar size needs to be provided.
R1.3: The pillars along with partitioning bars were inherited by ref 39. The elliptic shape was obtained by preliminary observation on the printing results where the circular shape does not ensure a faithful reproduction of the CAD geometries. Therefore, in this article we only compare pillars printed through a glossy or matte surface finished types, without a tedious reporting on the preliminary test on the geometries, and, hence, focusing on analyzing which dimensions best fitted the CAD values. An integration to the discussion on the pillar and, in general, on the accuracy of the geometries for the microfluidics results was provided at lines 200-204.
4) In figure 5, the full optical photograph of chamber device is suggested to be provided for better presentation of the separation performance from inlet to outlet.
R1.4: The full optical images were used instead of the previous ones as suggested by the referee for fig. 5. Line 230: substitution with the full optical photograph of the chamber device.
5) In section 3.2, as for the flow confinement test, the flow rate range tested is pretty narrow, the authors need to examine more higher flow rates to demonstrate the rapid analysis.
R1.5: The referee highlights a precise choice of Authors to use these narrow parameters instead of a wide range and an explanation can be summarized in the word optimization. Indeed, we obtain good performance of the device only in this very narrow range and larger ones were discarded by previous preliminary experiments. In details the text was revised as follows:
Line 225. “. In order to perform µFFE tests within 10 minutes, the best performance for..”
The point is that we want to figure out µFFE tests within 10 minutes, so as to be competitive with respect to commercially available kits such as Capturem™ Exosome Isolation Kit (Cell Culture) (https://www.takarabio.com/learning-centers/cell-biology/technical-notes/exosome-isolation-from-cell-culture) and ExoQuick-TC PLUS (https://systembio.com/shop/exoquick-tc-plus-exosome-purification-kit-tissue-culture-media/). We did not evaluate flow rates minor than 5 µL/min, both considering the time constrain and the fact that biological samples may denature if they are exposed to the electric field for a long period of time.
6) In this study, the authors used a relative simple model: only two different-sized polystyrene particles with similar surface charge. However, the situation is indeed more complicated in real biological settings. To make this study more attractive, it is recommended to add one test using at least three different-sized micro/nanoparticles or different surface-charged particles or real biological samples.
R1.6: Thanks to the referee suggestion and Editorial deadline extension, we managed to test bio-vesicles, i.e. exosomes (EXs) from Fetal Bovine Serum. These results have been added to the paper and properly discussed. We observed a difference in deviation with respect to NPs (with very similar Z-potential) and also a deformation due to the applied high voltage. In details:
Lines 287-301report biological tests performed with exosomes and a mixed population of exosomes and NPs. EXs and EXs/NPs collected at the outlets were characterized through nanoparticles tracking analysis. This technique allows determining simultaneously particles size distribution and concentration in a particles range between 10 nm and 2000 nm.
7) Lines 70-81, the authors mentioned the separation of exosomes, but they did not provide any direct tests. If the authors can demonstrate the separation of biological objects (e.g., cells and exosomes) using their 3D-printed electrophoresis device, this study will be very interesting.
R1.7: As reported in the previous answer, new results have been added to the manuscript and properly discussed.
Lines 287-301 report biological tests performed with exosomes and a mixed population of exosomes and NPs.
Minor comments:
1) Line 100 , “fittings (Fig. 1a,c). was directly”, correct this sentence.
R1.8: Line 107 - accomplished
2) Lines 155 and 156, “a volume of 60 L (100% concentrated solution) was loaded in a 555 L volume”, clarify the unit of volume.
R1.9: line 168-169 they are microliters, some symbols were lost. Accomplished
Reviewer 2 Report
This manuscript presents an inexpensive 3D-printed platform for micro- and nano-particle (M/NP) separation by free flow electrophoresis. The separation is characterized with two different anionic polystyrene beads (500 and 4000 nm diameters). The manuscript in generally well written, but a bit more experimental detail and additional discussion/analysis could strengthen the paper. Below are suggestions for minor revisions.
- A description of the "thermal process" (page 3 line 111) for the bonding of the PMMA lid should be included. Also, did the author's measure the effect of the thermal process on the pillar structures?
- It would be interesting to compare the measure zeta potential for the particles used to the observed electrophoretic mobility in FFE to see how well the two parameters align. If the voltage field is not known (the text only mentions the absolute magnitude of the voltage applied), a COMSOL simulation could probably be used calculate the field across the channel.
- The comparison of glossy vs. matte surfaces on page 6 is interesting, and the conclusion to use glossy appears to be the correct choice, but the use of percentages in the text (and in the abstract as 5% and conclusion as 5.5?) is confusing and perhaps not helpful. It appears that the 7% and 18% variations mentioned on lines 188-189 are simple averages of the pillar dimension uncertainties reported in Table I. This treatment appears to over-represent the uncertainty in pillar height and over-represent it for width. Also, the table should have a footnote reporting the statistical details ("uncertainties are standard deviations of x trials with y chips").
- The flow confinement discussion and figure on page 7 is a good addition, but I don't think trying only 1:1, 2:1, and 1:2 rations of buffer and analyte constitutes "optimization".
- Figures 6 and 7 are a little confusing. Why does one use % and the other concentration? (i) For Figure 6, I would suggest providing the size and charge of MP and NP, define DE% (or change to % of particles observed?) in the caption, and making it more clear why the voltages are different in panel b and why data for NPs are included in panel b but not panel a. (ii) For Fig 7, could you also report the plots (maybe for supplemental) for collection in channels 2 and 3? The Fig 6 data for 30 V in panel b are surprising, and it would be nice to see the overall trend in each channel (up to 40 V). Is it possible some NP just get "trapped" behind a pillar? Is this consistent with the slightly lower recovery of 84% NP reported on line 263?
Author Response
A description of the "thermal process" (page 3 line 111) for the bonding of the PMMA lid should be included. Also, did the author's measure the effect of the thermal process on the pillar structures?
R2.1: we want to thank the referee to highlight this point. We find that the term “thermal process” was misleading since this is a resin curing as described after. We revised the text to clarify the process at line 118 as follows:
“to achieve a uniform and irreversible bonding, the 3D printed part was sealed with the cover using as a glue the Poly(ethylene glycol) diacrylate (PEGDA) 575 (Sigma-Aldrich) mixed with 1% IRGACURE 819 (Sigma-Aldrich). The bonding was achieved by clamping the whole structure inside an aluminium frame and baking for 20 minutes on a hot plate at 120°C to obtain the full curing of the resin.”
It would be interesting to compare the measure zeta potential for the particles used to the observed electrophoretic mobility in FFE to see how well the two parameters align. If the voltage field is not known (the text only mentions the absolute magnitude of the voltage applied), a COMSOL simulation could probably be used calculate the field across the channel.
R2.2: We want to thank the referee for the suggestion and the idea for the model, which will be a reminder for future work. Indeed, the goal of this article is to investigate through an experimental approach the μFFE application, using 3D printed chip, on a variety of M/NPs and finally on bio-vesicles. For these reasons, we concentrated our effort, for the paper revision, to obtain new results on bio-vesicles, which were missing in the previous version. Indeed, thanks to the referee and Editorial deadline extension, we managed to test bio-vesicles, i.e. exosomes (EXs) from Fetal Bovine Serum. These new results have been added to the manuscript and properly discussed. We observed a difference in deviation with respect to NPs (with very similar Z-potential) and also a deformation due to the applied high voltage.
In details:
Lines 287-301 report biological tests performed with exosomes and a mixed population of exosomes and NPs. EXs and EXs/NPs collected at the outlets were characterized through nanoparticles tracking analysis. This technique allows determining simultaneously particles size distribution and concentration in a particles range between 10 nm and 2000 nm.
Therefore, a virtual study comprehensive of such a variety of analytes is highly complicated to setup. Besides the determination of the size and the zeta potential, the DLS instrument calculates the samples electrophoretic mobility and the conductivities. Values are reported in the following table, reported in the of the SI of the manuscript:
Table 3. DLS particles measurement
|
Sample |
Electrophoretic mobility [µmcm/Vs] |
Conductivity [mS/cm] |
|
MPs |
14.9 0.7 |
-0.9 0.2 |
|
NPs |
3.9 0.6 |
-1.0 0.8 |
|
EXs |
13.6 0.9 |
-0.7 0.1 |
Furthermore, we also considered studying the μFFE application through an analytical approach, but again the effort to obtain significant results for our final application exceed our actual objective.
The comparison of glossy vs. matte surfaces on page 6 is interesting, and the conclusion to use glossy appears to be the correct choice, but the use of percentages in the text (and in the abstract as 5% and conclusion as 5.5?) is confusing and perhaps not helpful. It appears that the 7% and 18% variations mentioned on lines 188-189 are simple averages of the pillar dimension uncertainties reported in Table I. This treatment appears to over-represent the uncertainty in pillar height and over-represent it for width. Also, the table should have a footnote reporting the statistical details ("uncertainties are standard deviations of x trials with y chips").
R2.3: we want to thank the referee for revision suggestion, the footnote reporting the statistical details has been added at line 215 and the % has been substituted with uncertainties
The flow confinement discussion and figure on page 7 is a good addition, but I don't think trying only 1:1, 2:1, and 1:2 rations of buffer and analyte constitutes "optimization".
R2.4: The Authors’ choice was to perform separation tests within 10 minutes using this technique and when we focused on this objective the right equilibrium was achieved after very few actually tested parameters. Indeed, also trying to investigate in a wider spectrum of parameters combinations, we obtained good performance of the device only in this very narrow range, so larger ones were discarded by very preliminary experiments. Saying that, to take into account the referee suggestion, the text was revised as follows:
Line 225. “. In order to perform µFFE tests within 10 minutes, the best performance for..”
The point is that we want to figure out µFFE tests within 10 minutes, so as to be competitive with respect to commercially available kits such as Capturem™ Exosome Isolation Kit (Cell Culture) (https://www.takarabio.com/learning-centers/cell-biology/technical-notes/exosome-isolation-from-cell-culture) and ExoQuick-TC PLUS (https://systembio.com/shop/exoquick-tc-plus-exosome-purification-kit-tissue-culture-media/). We did not evaluate flow rates minor than 5 µL/min, both considering the time constrain and the fact that biological samples may denature if they are exposed to the electric field for a long period of time.
Figures 6 and 7 are a little confusing. Why does one use % and the other concentration? (i) For Figure 6, I would suggest providing the size and charge of MP and NP, define DE% (or change to % of particles observed?) in the caption, and making it more clear why the voltages are different in panel b and why data for NPs are included in panel b but not panel a. (ii) For Fig 7, could you also report the plots (maybe for supplemental) for collection in channels 2 and 3? The Fig 6 data for 30 V in panel b are surprising, and it would be nice to see the overall trend in each channel (up to 40 V). Is it possible some NP just get "trapped" behind a pillar? Is this consistent with the slightly lower recovery of 84% NP reported on line 263?
R2.5: (i) Following the suggestion of the referee we changed the DE% in % of particles observed and, since preliminary μFFE tests were performed with MPs it was not necessary to select a narrower range of voltages, but we focused on a range between 0 and 40 V to evaluate where MPs would be collected at the outlets. A finer range of voltages was tested with a mixed population of MPs and NPs to highlight the different distribution of MPs and NPs at the outlets. (ii) Then, we added in the SI plots of particles collected at outlet#2 and outlet#3 as depicted for Fig.7. Fig 6 in panel b does not report MPs and NPs at 0 V, since the overall of MPs and NPs are collected at outlet#3. In panel b is reported a narrower range of voltages since a different distribution of particles collected at the outlets results to be significant for that range. Yes, it is possible that some NPs can remain trapped behind the pillars, but just a very little fraction, while a major “leak” is close to the electrodes, as we hypothesized, “.. a little amount of particles passed over the partition bars toward the +V electrode channel. ” (line 280)
Round 2
Reviewer 1 Report
Most of my previous questions have been properly solved. And one more suggestion for the authors: the repeat times of their tests need to be clarified, and the error bars should be added in figures like Figures 6, 8, and 9.
Author Response
We want to thank you for the suggestion. The number of experiments for each condition have been clarify and the error bars inserted.
At line 236 pag 8 the text was revised and the figures modified
For each set of tested analytes, the experiments in the μFFE device were repeated three times, error bars were reported according to the acquired data over the repetitions.